# Impact of school SES on literacy development

**Naymé Salas**[1]*, **Mariona Pascual**[1,2]

1 Departament de Didàctica de la Llengua i la Literatura, i de les Ciències Socials, Facultat de Ciències de l'Educació, Universitat Autònoma de Barcelona, Barcelona, Spain, 2 Centro de Investigação em Ciência Psicológica (CICPSI), Faculdade de Psicologia, Universidade de Lisboa, Lisboa, Portugal

* nayme.salas@uab.cat

## Abstract

Effective literacy skills are essential to actively participate in today's society. However, little research has been conducted that examined the impact of contextual variables on literacy development. The present paper addressed whether and how the socioeconomic status of the school (S-SES) children attend affects their literacy achievements. Eight-hundred and seventy-eight 2nd and 4th grade children participated in the study. Data were collected in low-SES (vulnerable) and in mid-high-SES (non-vulnerable) schools. Children completed a large battery of language, cognitive, and literacy tasks in Catalan, a language spoken in a region in Spain where virtually all children are at least bilingual (they also speak Spanish) and it is the main language of instruction. Results showed that children in vulnerable schools were outperformed by children in non-vulnerable schools across all literacy competencies, but particularly affected higher order skills; that is, text quality and reading comprehension. Differences with their non-vulnerable peers remained, even after controlling for context-level covariates, including familial SES. However, S-SES ceased to exert significant influence once children's cognitive and, especially, linguistic skills were considered. The study adds to previous research claiming that school SES has an effect on students' literacy skills, above and beyond children's home SES. However, our findings also suggest that literacy performance is ultimately mostly dependent on educationally actionable, subject-level skills. Educational implications are discussed.

## Introduction

Academic and professional success are highly dependent on learning to read and write effectively [1, 2]. Moreover, competent reading and writing are not only necessary to understand and to communicate ideas at school and in professional environments, but they are both powerful learning tools [3]. Learning to read and to write are, thus, the cornerstones on which equality of opportunity rests. In recent decades, the role of subject-level factors (e.g., oral language, cognitive skills) on literacy development has been investigated extensively [4–7]. However, comparatively little research has been conducted that examined the impact of contextual variables on literacy, particularly in non-English-speaking contexts. The present paper addressed whether and how children's socioeconomic environment affects their achievement in literacy competencies.

**Data Availability Statement:** Data contains potentially sensitive information of underaged children. We did not obtain informed consent from parents to make these data available, even if it has been confidentially treated. Besides requesting it to both authors, data can be requested via email to

the support unit of the Comisión de Ética en la Experimentación Animal y Humana (CEEAH), from the Autonomous University of Barcelona, at uisad. c.educacio.academic@uab.cat

**Funding:** This research was supported by grants 2015ACUP 00175 (Recercaixa-ACUP program, https://caixaresearch.org/es/convocatoria-caixaresearch-recercaixa-investigacion), PID2019-108791GA-I00 (Ministerio de Ciencia Innovación y Universidades, https://www.ciencia.gob.es), and 2021-SGR-00205 (AGAUR support to research groups) awarded to N.S. There was no additional external funding received for this study. The funders had no role in study design, data collection and analysis, decision to publish, or preparation of the manuscript.

**Competing interests:** The authors have declared that no competing interests exist.

## Socioeconomic status' effects on learning to read and write

A child's socioeconomic status (SES) is a measure that results from a combination of three main aspects: parental education, income or material resources at the household, and parental occupation [8, 9]. Often, a measure of one or more of these aspects is used to determine its effect on development or behavior [10, 11]. Previous meta-analytic studies have consistently found that a child's SES has a relatively small, but consistent and significant impact on their development, including language and literacy [12–14]. The mechanisms through which SES and literacy achievement are related are not entirely clear [9]. The nature of parent-child factors may be partly responsible for children's literacy outcomes [10, 15]. In this sense, some studies have pointed out that parental education is more important than their occupation or income to predict children's academic achievement [16], probably because children are typically exposed to reading and writing practices more often and from birth in households with more educated adults; and more exposure has been found to lead to better outcomes [17]. In any case, there is abundant evidence that children's early literacy experiences, which are highly influenced by familial SES, set the stage for their subsequent literacy development [18]. Importantly, children are not only sensitive to the SES of their familial context. A number of large-scale studies have provided compelling evidence that the SES of the school children attend has a substantial impact on literacy development above and beyond the effect of child-level SES [13, 19, 20]. For example, the 2010 OECD report for literacy achievements indicated that not only was school SES relevant to explaining findings, but it was found to be even more strongly related to reading outcomes than individual SES [8]. There does not seem to be a consensus over what are the drivers of differing academic performance as a function of school SES. School composition has been proposed as a key factor in academic achievement in several studies. In most of these studies, school SES has been operationalized as the average of students' individual SES [13, 21–24]. Moreover, some studies point to the fact that school climate, teacher practices and attitudes, as well as emphasis on academic achievement may be found to differ in low-SES schools when compared to more affluent schools [25]. In short, SES at the individual but, especially, at the school level has been shown to significantly affect literacy outcomes [24–26]. Nonetheless, most research has been conducted in English-speaking populations and has focused on overall academic achievement, rather than on specific literacy skills. Given the importance attributed to school SES in past research, the present paper intended to examine its role in reading, as well as in writing achievements in primary school in a non-English-speaking context: Catalonia (Spain).

## Key literacy skills

Reading and writing allow us to understand and interact with others in the social context in which we live [27–29]. Reading is the ability to (re)construct the meaning of a text, by generating a mental representation of what the text says, and to integrate such representation with previous knowledge [30]. Reading thus involves the execution of word-level processes, chiefly, word reading, as well as text-level processes, such as comprehension monitoring or inference resolution. These processes are supported by domain-general cognitive resources such as working memory or inhibitory control [31], while they require strong orthographic and linguistic representations at the phonological, morphological, syntactic, and discursive-pragmatic levels [32, 33]. Writing is a complex skill that involves transforming ideas into linguistic formulations, which are transposed into scripted material that serves a communicative function [27, 34, 35]. Writers engage in a series of recursive domain-specific processes, including generating and organizing ideas, setting themselves rhetorical goals, searching for relevant content, as well as revising text [36–38]. An important milestone in writing development is the

automatization of transcription skills, that is, spelling and handwriting; otherwise, these mechanical aspects of writing overload working memory resources, resulting in poor texts [39]. Similarly to reading, research has accrued that writing competence at the sub-lexical (i.e., handwriting), lexical (i.e., spelling), and text levels rests on an individual's domain-general cognitive resources, e.g., working memory [40], and linguistic representations, e.g., morpho-syntactic skills, vocabulary [33]. In order to ascertain the specific impact of school SES, we collected estimates of important predictors of reading and writing, including general non-verbal intelligence, other domain-general cognitive skills (working memory, inhibition), as well Rapid Associative Naming (RAN), a measure of processing speed, because of its known impact on word- and text-level literacy across languages [4, 41, 42]. Additionally, we obtained measures of children's linguistic skills (grammar and vocabulary). These measures were used as covariates to estimate S-SES impact on literacy outcomes above and beyond known predictors of literacy competence.

## This study

The main aim of the present study was to examine the impact of school SES (henceforth, S-SES) on children's literacy achievements. We were interested in how S-SES affected grade 2 and grade 4 students learning to read and write in Catalan. Consequently, we collected data in vulnerable and more affluent public schools. Crucially, "vulnerable schools" were regarded as such using the criteria and classification of the Catalan Department of Education (more details in Method), rather than computing the average of children's individual SES in each school. At the participating schools, we administered tasks to estimate children's reading (e.g., word identification, reading comprehension) and writing (e.g., spelling, writing productivity, and writing quality) skills and conducted MANCOVA tests to determine whether S-SES had an impact on such skills. Across analyses we controlled for demographic factors: children's gender and age. Moreover, we investigated whether S-SES effects were modified when we also controlled for contextual factors: child-level SES (henceforth, C-SES) and children's exposure to the language of instruction outside school (Catalan). Finally, we examined whether S-SES effects remained after also controlling for a host of subject-level factors (e.g., IQ, vocabulary, working memory) known to affect literacy development. The study thus improves on previous research by (1) addressing an under-researched, non-English-speaking context; (2) comparing schools on the basis of external, objective assessments of SES composition, rather than simply averaging the individual SES of children in classrooms; (3) controlling for a large number of demographic, contextual, and individual-level factors with a substantial record of impacting literacy performance; and (4) considering the full range of literacy skills, both reading and writing, at the (sub)word- and text-levels. Our research questions (RQs) were, thus,

RQ1: Does school SES affect literacy achievement in 2nd- and 4th-grade students above and beyond the effect of individual SES?

RQ2: Which literacy skills (e.g., handwriting, reading comprehension) are most sensitive to S-SES?

RQ3: Does school SES have an impact on literacy achievements above and beyond the effect of known predictors of reading and writing performance?

With regards to RQ1, we expected that, in line with previous studies, school SES would make a significant impact on literacy scores even after controlling for demographic variables and the child's individual SES. Given that previous research has found that both reading and writing are sensitive to SES differences [13, 43], we expected an impact of S-SES across all

literacy skills. As for RQ3, we were ambivalent as to the effect of S-SES on literacy skills after controlling for known predictors of performance since, to the best of our knowledge, no studies have addressed this issue to the extent that we have; that is, including the most relevant cognitive and linguistic underpinnings of reading and writing, in addition to demographic and contextual variables. Because individual SES and S-SES have been found to have a consistent but only small-to-moderate effect on literacy outcomes [12], it is possible that the influence of S-SES can no longer be observed after controlling for highly stable literacy-related factors. Nevertheless, we believe that, given that we have used a different means to classify schools as vulnerable, where low-SES overlapped with other risk factors (e.g., low to null exposure to the language of instruction), S-SES might still have an effect above-and-beyond our chosen demographic, contextual, and subject-level covariates.

## The Catalan context

This study was carried out in Catalonia, a region of Spain where the population is typically bilingual (Catalan-Spanish). Catalan has a semi-opaque orthography [44–46], more consistent than English, but less so than Spanish. It is morphosyntactically similar to Spanish, as they are both Latin-rooted Romance languages. Importantly, Catalan is the language of instruction across the public school system in Catalonia. Nonetheless, not all children have the same degree of exposure to Catalan outside school. Some children (e.g., with an immigrant background) are usually not exposed to Catalan outside of the school environment [47, 48]. Moreover, children in disadvantaged areas, who are typically not exposed to the language of instruction for everyday communication, may take longer to develop linguistic skills in Catalan [49]. Because the particular context of the study may affect the impact of both C-SES and S-SES on literacy outcomes, we obtained information about children's exposure to the language of instruction.

## Materials and methods

### Participants

The parents or legal guardians of 436 2nd graders and 437 4th graders gave explicit, written consent to their children participating in the study (N = 873). All data were collected in compliance with the guidelines of the Ethics Commission for Animal and Human Experimentation (CEEAH) of the Universitat Autònoma de Barcelona. Table 1 shows basic demographic information for all participants. Children were recruited in socioeconomically vulnerable and non-vulnerable schools in Barcelona and neighboring cities. The Catalan Department of Education applies a set of criteria to identify each type of school. Schools with a high proportion of students from low-income households, who typically do not speak the language of instruction

**Table 1. Demographic characteristics and distribution of participants by Grade and School Type.**

|  | Non-vulnerable schools | | Vulnerable schools | |
| --- | --- | --- | --- | --- |
|  | Grade 2 | Grade 4 | Grade 2 | Grade 4 |
|  | M (SD) | M (SD) | M (SD) | M (SD) |
| *N (boys)* | 268 (140) | 279 (149) | 168 (70) | 158 (88) |
| Mean age | 8;8 (0;3) | 10;10 (0;4) | 8;8 (0;4) | 10;10 (0;3) |
| ISEI Score | 54.45 (13.86) | 53.38 (14.29) | 39.29 (12.09) | 37.97 (9.97) |
| Catalan Exposure | 1.29 (1.06) | 1.33 (1.05) | 0.27 (0.54) | 0.21 (0-47) |

*Note.* ISEI = International Socio-economic Index Score [51], range 20—90: Catalan Exposure range (0-2.5).

outside of the school, and whose parents generally have low educational levels are labeled "high-complexity" educational centers (henceforth, "vulnerable" schools, VSs). On the other hand, regular or "non-vulnerable" schools (NVSs) are those where children usually come from mid- to high-income backgrounds, whose parents have often completed higher education studies, and who are frequently exposed to Catalan outside the school setting. Upon identification, vulnerable schools are eligible to receive more resources than other schools, including a higher ratio of teachers per student and a more diverse body of professionals to attend to students' needs [50]. In our study, we compared literacy attainment in "vulnerable" schools to "non-vulnerable" schools, using the classification by the Catalan Department of Education to ascertain the role of S-SES.

Gender distribution was similar across school types, $Chi^2(1) = 1.56$, $p = .212$, and between grades, $Chi^2(1) = 1.07$, $p = .302$. We administered a sociolinguistic questionnaire to the children, with the assistance and verification of the teachers. Such a procedure was preferred to directly addressing students' parents because of access issues, and because a large proportion of the sample included children with illiterate parents who, often, did not speak Catalan or Spanish. The main aim of the questionnaire was to determine parents' occupation and educational level, as well as exposure to Catalan outside the school setting.

**Child-level SES.** We used the International Socioeconomic Index (ISEI) by Ganzeboom et al. [51], which allowed us to assign a score on a continuous scale according to occupation. We inquired about the occupations of the child's parents or, alternatively, of their legal guardian. Although the research by Ganzeboom et al. [51] only includes occupation-ISEI score matches for men, we obtained ISEI scores also for women parents and used the average of both parents as the final variable of C-SES, a procedure that has proven reliable in previous studies [52]. A two-way ANOVA showed that the average ISEI score (i.e., child-level SES) of children in VSs was significantly lower than that of children attending NVSs, $F(1, 759) = 242.94$, $p < .001$, $\eta^2 = .242$. There were no significant differences between grades, $F(1, 759) = 1.50$, $p = .221$, and there was no Grade by School Type interaction, $F < 1, p = .900$.

**Exposure to Catalan.** We also used the sociolinguistic questionnaire to inquire about children's exposure to the language of instruction (Catalan) outside of the school setting. Specifically, we asked each child which language(s) they used to talk to their parents, siblings, and friends. When Catalan was the only language used to speak to either parent (or the legal guardian), we assigned a score of 1 to each parent. In cases of single-parent families, the score was doubled; that is, when Catalan was the chief means of communication within a monoparental family, we assigned a score of 2. If a child typically used either of two languages (e.g., Catalan or Spanish) to talk to one of their parents, a score of 0.5 was assigned to that parent. If a child would use three possible languages with a parent (e.g., Catalan, Spanish, German), that parent was assigned a score of 0.33. When children reported using Catalan to speak to friends or siblings, we added 0.25 points to the final score for each type of interlocutor (friends, siblings). The final score thus ranged between 0 to 2.5. Children in VSs were significantly less exposed to Catalan outside school than children in NVSs, $F(1, 807) = 263.83$, $p < .001$, $\eta^2 = .247$, and this exposure was not different as a function of Grade, $F < 1$, $p = .923$, while there were no significant interactions, $F < 1$, $p < .461$.

## Tasks and measures

We administered a series of standardized and mostly adapted or bespoke instruments for collecting data on children literacy competence and related skills. The main reason behind the use of non-standardized instruments was that only reading comprehension tests are available that have been standardized for Catalan-speaking school-aged children. Whenever possible,

we used adaptations of tests and subtests standardized for Spanish (e.g., Raven's matrices, to measure non-verbal IQ; or the Digits subtest from the WISC-IV test battery [61, 64]). In the section below, we provide details of the adaptations that have been made and their underlying criteria. We are certain that validity was not an issue with any adaptation nor with bespoke tasksm since they have all been used in previous studies for virtually identical purposes (e.g., alphabet task to measure handwriting fluency [53]). However, we report our own reliability calculations to determine the degree to which the adapted or researcher-designed instruments were reliable.

**Handwriting fluency.** We used this test to assess handwriting skills. Children wrote the alphabet as fast as possible in 60 seconds [53]. The final score was the sum of all correctly written letters that they were able to produce in the first 15 seconds. In order not to penalize children in centers where the alphabet was not taught [54], order of the letters was not taken into account. Inter-rater reliability on a random 20% of the sample was of.984 (ICC).

**Spelling accuracy.** To assess spelling, children were asked to write 34 words from a bespoke dictation task [55]. Words were presented in one of two fixed, pseudo-randomized orders, which were counterbalanced. Items in the dictation task were also counterbalances for frequency, length in syllables, and syllabic complexity. Each word was delivered out loud in isolation, contextualized using a carrier sentence, and then repeated in isolation again. The final score was the sum of all correctly written words. Inter-rater reliability on a random 20% of tasks was.989 (ICC). The task's internal consistency had Cronbach's $\alpha$ of.79.

**Text writing.** Each child produced a narrative and an argumentative text (opinion essay). There were two prompts for each genre, which were administered randomly to each classroom. The opinion-essay prompts were (1) *Do you think that all children your age should go to school?* or (2) *Do you think that there should be recess time at school?*. The narrative prompts were (1) *Write a story about a child who has lost his/her pet* or (2) *Write a story about a child who is angry at his/her friend*. A research assistant presented the prompt and gave children up to 10 minutes to plan their text and then up to 20 minutes to write the text. Both text samples were transcribed by two trained research assistants in CHAT-compatible format [56]. Spelling mistakes were corrected and punctuation was minimally revised or corrected to facilitate the readability of the texts. Transcription fidelity between research assistants was.996 (ICC). From these transcriptions, a number of text-based measures were obtained: (1) the total number of words, to estimate text productivity, was automatically counted with CLAN [55]. All words were counted, regardless of spelling mistakes. Also, (2) texts were evaluated for holistic text quality by at least two independent raters. Ratings ranged from 0 (considerably below grade-level expectations) to 5 (considerably above grade-level expectations). Grade-level expectation followed the milestones and requirements established in the Primary Education curriculum of the Catalan Government for each grade level (Decree 119/2015) [57]. Raters were trained for a specific grade level (2nd or 4th) to ensure consistency and age-appropriate assessment. In cases where raters had a discrepancy of more than 1 point, a third independent rater assessed the text. The final score was the average between the two (or three) ratings. Raters used the Text Handler app (for further details on the procedure and training, see [58]). Interrater reliability (ICC) was.857, for grade 2, and.725, for grade 4.

**Word reading.** Children were administered the word reading subtest of the PROLEC-R battery [59]. The test invites children to read 40 words aloud of varying complexity. The number of words read correctly is tallied and the time is registered. The score is the number of words accurately read in one minute. Reported reliability in the manual is Cronbach's $\alpha$ = .79.

**Reading comprehension.** We administered an adapted version of the ACL: *Avaluació de la Comprensió Lectora* 'Reading comprehension Evaluation' test [60]. The test includes a collection of brief texts of various genres, with multiple-choice questions to measure children's

understanding of literal information, inferences, and their global understanding of the text. The manuals provide a different instrument for each grade level (from 1st through to 6th). Test instructions indicate that 2nd grade children are administered the 1st-grade instrument (3rd-grade children are administered the 2nd-grade instruments, and so on), especially if tested during the first trimester of the school year (as was our case), in order to determine that they have acquired grade-level reading comprehension competences. To prevent ceiling-effects, we administered a slightly longer version of the 1st-grade and 3rd-grade instruments (to our 2nd- and 4th-grade participants, respectively), where we included two more items obtained from the 2nd- and 4th-grade instruments, respectively. Children had up to 45 minutes to complete the test. The score was the total proportion of correct responses. Reliability (Chronbach's $\alpha$) was.816, for the 2nd-grade instrument, and.898, for the 4th-grade instrument.

**Non-verbal IQ.** Non-verbal intelligence was assessed with the Progressive Raven Matrices test [61]. All children completed the five sets of 12 tasks in 45 minutes. The manual reports a reliability of.86 (Cronbach's $\alpha$).

**Rapid Associative Naming (RAN).** To assess speed of processing, children were administered a RAN-letters task [62]. Children were asked to name five letters of the alphabet (a, s, d, p, o), displayed randomly on an A4 sheet 40 times (in five rows of eight letters each), as fast as they could. There was one practice trial, only to ensure children were familiar with the letters and their names, and two test trials. The final score was the mean time to complete each trial. Reliability was the correlation between test trials, $r = .834$.

**Inhibition.** To assess the executive function of inhibition, children were administered the *Same-word, Opposite world test* [63]. In the "same world" trials, children had to say outloud the numbers '1' or '2' printed on a path, as fast as possible. After a same-world, baseline trial, two-opposite world test trials ensued, where children were asked to say ther number opposite to the one printed on the path (e.g., say "one" if they saw number "2"), as fast as they could; that is, they had to inhibit a prepotent response. The final score was the mean time across the two opposite-world trials. Reliability was the correlation between test trials, $r = .823$.

**Working memory.** To assess working memory we used the Digits subtest of the Spanish adaptation of the WISC-IV test battery [64]. Children had to recall lists of digits that increased from two to eight in length. Two practice items preceded test items in both the forward (i.e., repeating digits in the same order as they are delivered) and backward (i.e., repeat digits in reverse order) conditions. As per the manual's instructions, the test was discontinued if two items within the same list-length were recalled incorrectly. Each correctly recalled string was awarded 1 point, and the overall score was the sum of both direct and reverse tasks. Reliability was.74 (Cronbach's $\alpha$).

**Vocabulary.** We used an adaptation of the Vocabulary subtest of the WISC-IV Spanish test battery [64] to assess Catalan vocabulary. Expert researchers and teachers selected items similar to the words in the Spanish version. We took into account, especially, context and frequency of use, age of acquisition, and difficulty. The final test consisted of 26 items of increasing difficulty, where children had to provide oral definitions. Scoring was based on depth and specificity as per the manual's instructions, and Spanish definitions were accepted. The final score was the sum of all points obtained. A trained RA rescored a random 16% of the sample, showing high consistency of scoring (ICC = .958). Internal consistency was Cronbach's $\alpha$ = .887.

**Morphosyntactic knowledge.** We assessed receptive morphosyntactic skills using an adaptation of a sentence-reading test of the PROLEC-R [59]. The administrator read a sentence out loud while the child was shown four alternative pictures. Children were asked to point to the picture that best matched the aurally-delivered sentence. The test had 16 items of

increasing difficulty. One point was given for each correct answer and the final score was the sum of all correct responses. Reliability was Cronbach's $\alpha$ = .83.

# Results

## Psychometric properties of variables, descriptive statistics, and analytical strategy

Table 2 shows the means and standard deviations of all outcome and predictor variables by grade and school type. Skewness and kurtosis values were assessed to detect abnormalities in the distribution of values. Absolute skewness values ranged between 0.13 and 2.82, while absolute kurtosis values ranged from 0.21 to 14.82, indicating a relatively normal distribution [65]. We computed z-scores for all outcome measures, for each grade separately. For the sake of parsimony, and to avoid highly-correlated dependent variables, we computed composite variables for the number of words variables and the text quality variables, using the mean of the standardized score for each genre (narrative and argumentative). Missing values were also assessed for all intervening variables. We investigated variables in which missing values exceeded 5%. These were C-SES (14.8%), Exposure to Catalan (9.5%), Spelling accuracy (8.3%), Word Reading (5.5%), and Reading Comprehension (8.1%). To determine whether the rate of missing values was the same across School Types (the variable of interest), we ran $Chi^2$ tests for 2x2

**Table 2. Grade and School Type.**

| | Non-vulnerable Schools | | Vulnerable Schools | |
|---|---|---|---|---|
| | Grade 2 | Grade 4 | Grade 2 | Grade 4 |
| | M (SD) | M (SD) | M (SD) | M (SD) |
| *Subject-level predictor variables* | | | | |
| Non-verbal IQ[1] | 27.18 (10.51) | 38.27 (8.36) | 24.58 (9.90) | 31.95 (10.79) |
| WM1 | 11.36 (2.20) | 12.73 (2.49) | 9.87 (2.25) | 11.45 (2.35) |
| Inhibition[2] | 52.59 (12.36) | 40.18 (7.96) | 53.24 (11.17) | 41.98 (9.07) |
| RAN[2] | 31.90 (10.38) | 22.12 (5.73) | 33.21 (8.78) | 22.70 (5.71) |
| Vocabulary[1] | 14.34 (4.98) | 19.38 (4.38) | 9.67 (4.28) | 14.76 (4.84) |
| MS skills[3] | 66.51 (17.78) | 74.11 (16.04) | 55.06 (16.02) | 67.74 (14.01) |
| *Outcome Literacy Variables* | | | | |
| Handwriting | 5.20 (1.95) | 8.40 (2.77) | 4.93 (1.84) | 7.88 (2.89) |
| Spelling[1] | 8.68 (6.19) | 13.85 (6.00) | 6.51 (5.72) | 11.37 (5.99) |
| Productivity OP[4] | 22.98 (17.08) | 46.69 (24.71) | 21.53 (15.11) | 41.62 (25.84) |
| Productivity NA[4] | 60.82 (35.88) | 113.76 (54.05) | 36.01 (31.71) | 86.97 (53.87) |
| Text quality OP[6] | 0.81 (0.60) | 0.83 (0.58) | 0.56 (0.55) | 0.63 (0.55) |
| Text quality NA[6] | 1.34 (0.69) | 1.47 (0.89) | 0.88 (0.69) | 0.97 (0.81) |
| Word reading[6] | 23.02 (15.96) | 43.55 (23.32) | 19.40 (15.96) | 39.19 (20.84) |
| Reading comprehension[3] | 63.43 (20.47) | 65.30 (21.17) | 48.44 (19.50) | 47.96 (22.03) |

*Note.* WM = working memory; MS = morphosyntactic; ISEI = International Socio-economic Index Score [51], range 20—90: Catalan Exposure range (0-2.5);

OP = opinion essay; NA = narrative text.

[1] Raw score

[2] In seconds

[3] Percentage correct

[4] Number of words

[5] z-score average of OP and NA texts

[6] range: 0-5

contingency tables for each of the variables with missing values above 5%. These tests indicated that for Catalan Exposure, Spelling Accuracy, the Number of Words composite variable, and Reading Comprehension, there were significantly more missing values in the vulnerable than in the non-vulnerable schools. These missing value rates are not uncommon in research on Education [66], while there are authors that claim that missing values under 10% should not bias results [67]. In this sense, our only variable with a larger percentage of missing values.

(C-SES) presented a similar missingness rate across school types ($p > .05$). However, readers should take into account the potential for bias in the findings to be reported next. Sample size was imbalanced across groups, with a larger number of students in the non-vulnerable schools than in the vulnerable ones. Although we attempted to recruit a similar number of students for both groups, classroom size is often slightly smaller in vulnerable schools to comply with teacher-student ratios that are mandatory in "high-complexity" schools (Resolució ENS/906/2014, 2014 [50]). To avoid bias in the analyses, we used the Weighted Least Square (WLS) option in SPSS, to have balanced (50/50) sample sizes. To answer our research questions we adopted a four-phase procedure. First, we conducted a one-way MANCOVA for all outcome measures, with School Type as the fixed factor and with Gender and Age as covariates. Given that we had standardized all values for each grade separately, there was no need to include Grade in any analyses. Second, we conducted a one-way MANCOVA, identical to the first, but we added the contextual variables of Exposure to Catalan and C-SES as covariates. Third, we ran a MANCOVA where we also included a number of variables that measured children's levels on a series of cognitive skills; specifically, non-verbal IQ, Rapid Associative Naming (RAN), inhibition, and working memory. Fourth, we added linguistic skills as covariates; specifically, receptive morphosyntax and vocabulary. Across MANCOVAs we assessed the impact of School Type on all literacy outcome measures: word reading, handwriting, spelling, reading comprehension, and text composition. We used MANCOVA particularly to avoid Type I errors resulting from conducting several tests on related, dependent variables [68]. Analyses were carried out using SPSS v.27. Box's test of equality of covariance matrices was significant across MANCOVAs. Therefore, we used Pillai's trace criterion to assess multivariate significance, since it is the most robust when this assumption is not met [68]. We compared effect sizes (partial eta-squared) for all significant School-Type effects across our phases of analysis. In order to obtain robust estimates, all analyses included a bootstrapping procedure of 2,000 samples.

## School SES effects on literacy achievement

Our first MANCOVA tested the main effect of School Type, while controlling for the effects of Gender and Age, which were entered as covariates into the model. Results showed that School Type had a large significant effect on the dependent variables, $F(6, 742) = 24.98$, $p < .001$, $\eta_p^2 = .168$. Age was non-significant, $F(6, 742) = 0.47$, $p = .835$, $\eta_p^2 = .004$, while Gender was a significant covariate, with a medium effect size, $F(6, 742) = 8.95$, $p < .001$, $\eta_p^2 = .068$, whereby girls outperformed boys. Between-subject analyses indicated that School Type had an effect on all outcome variables, which favored children in non-vulnerable schools. The School Type effect was thus statistically significant and of medium size for handwriting fluency, $F(1) = 8.31$, $p = .004$, $\eta_p^2 = .011$, spelling accuracy, $F(1) = 21.50$, $p < .001$, $\eta_p^2 = .028$, text productivity, $F(1) = 27.51$, $p < .001$, $\eta_p^2 = .036$, and word reading, $F(1) = 11.21$, $p < .001$, $\eta_p^2 = .015$. The impact of School Type on text quality, $F(1) = 85.27$, $p < .001$, $\eta_p^2 = .102$, and reading comprehension, $F(1) = 117.48$, $p < .001$, $\eta_p^2 = .136$, was also significant and of a considerably larger size. In short, after controlling for basic demographic variables, children attending vulnerable schools showed a poorer performance across all literacy skills. Our second MANCOVA included, in

addition to Gender and Age, C-SES and Exposure to Catalan, as context-level control variables. This analysis also revealed a significant effect of School Type on the dependent variables, albeit of reduced size, $F(6, 625) = 4.61$, $p < .001$, $\eta_p^2 = .042$. Age remained a non-significant factor, $F(6, 625) = 0.45$, $p = .843$, $\eta_p^2 = .004$, while Gender remained a significant covariate, $F(6, 625) = 7.93$, $p < .001$, $\eta_p^2 = .071$. The two additional, context-level covariates had a significant impact on literacy achievement levels: both Exposure to Catalan, $F(6, 625) = 3.95$, $p < .001$, $\eta_p^2 = .036$, and C-SES, $F(6, 625) = 7.24$, $p < .001$, $\eta_p^2 = .065$, were positively associated with higher scores; in other words, more exposure to the language of instruction outside school and a higher SES family background were each associated with higher literacy performance scores, regardless of the type of school children attended. After controlling for demographic and context-level variables, the School Type effect was no longer significant for handwriting fluency, $F(1) = 0.46$, $p = .498$, $\eta_p^2 = .001$, spelling accuracy, $F(1) = 1.12$, $p = .291$, $\eta_p^2 = .002$, text productivity, $F(1) = 2.92$, $p = .088$, $\eta_p^2 = .005$, or word reading, $F(1) = 2.83$, $p = .093$, $\eta_p^2 = .004$. However, children in vulnerable schools were still outscored by children in non-vulnerable schools in text quality, $F(1) = 13.64$, $p < .001$, $\eta_p^2 = .021$, and reading comprehension, $F(1) = 12.22$, $p < .001$, $\eta_p^2 = .025$. All in all, attending a vulnerable school resulted in a disadvantage for children's literacy skills, particularly for the development of text composition and comprehension, above and beyond the effect of crucial demographic and context-level variables. Our third MANCOVA tested the effect of School Type, controlling for demographic and contextual variables, while adding relevant cognitive skills as covariates. School Type remained a significant factor, although the effect size was small, $F(6, 590) = 2.73$, $p = .013$, $\eta_p^2 = .027$. Age did not have a significant effect on literacy outcomes, $F(6, 590) = 0.28$, $p = .946$, $\eta_p^2 = .003$, while Gender did, $F(6, 590) = 7.88$, $p < .001$, $\eta_p^2 = .074$. Although children's exposure to Catalan remained a significant factor explaining literacy performance, $F(6, 590) = 4.51$, $p < .001$, $\eta_p^2 = .044$, C-SES was no longer significant, $F(6, 590) = 1.34$, $p = .236$, $\eta_p^2 = .013$. All cognitive-skill variables had a significant effect on literacy outcomes. The largest one was RAN, $F(6, 590) = 19.67$, $p < .001$, $\eta_p^2 = .167$, followed by non-verbal IQ, $F(6, 590) = 15.24$, $p < .001$, $\eta_p^2 = .134$, and working memory, $F(6, 590) = 12.53$, $p < .001$, $\eta_p^2 = .113$. Finally, inhibition also had a small-size effect on literacy outcomes, $F(6, 590) = 3.89$, $p < .001$, $\eta_p^2 = .038$. Between-subject analyses showed that the School Type effect was only significant for text quality, $F(1) = 7.99$, $p = .005$, $\eta_p^2 = .013$, and reading comprehension, $F(1) = 6.94$, $p = .009$, $\eta_p^2 = .009$; however, it was non-significant for handwriting fluency, $F(1) = 1.43$, $p = .233$, $\eta_p^2 = .002$; spelling, $F(1) = 0.04$, $p = .847$, $\eta_p^2 < .001$; text productivity, $F(1) = 1.47$, $p = .226$, $\eta_p^2 = .002$; or word reading, $F(1) = 1.38$, $p = .240$, $\eta_p^2 = .002$. Finally, in our fourth MANCOVA we tested the School Type effect when adding oral-language skills (receptive morphosyntax and vocabulary) to our host of covariates. Analyses showed the School Type effect was no longer significant, $F(6, 587) = 0.83$, $p = .548$, $\eta_p^2 = .008$. As in previous analyses, Gender had a significant effect on literacy outcomes, $F(6, 587) = 9.78$, $p < .001$, $\eta_p^2 = .091$, while Age did not, $F(6, 587) = 0.348$, $p = .917$, $\eta_p^2 = .003$. C-SES did not have an effect on literacy performance over and above the other variables in the model, $F(6, 587) = 0.88$, $p = .507$, $\eta_p^2 = .003$, but children's exposure to the language of instruction continued to exert significant influence on literacy skills, $F(6, 587) = 3.67$, $p = .001$, $\eta_p^2 = .036$. Cognitive covariates maintained their significant influence on literacy outcomes, although some were reduced in size. This was the case for non-verbal IQ, $F(6, 590) = 10.07$, $p < .001$, $\eta_p^2 = .043$, and working memory, $F(6, 590) = 9.45$, $p < .001$, $\eta_p^2 = .088$. Inhibition had an effect virtually identical to the previous analysis, $F(6, 590) = 3.88$, $p < .001$, $\eta_p^2 = .038$, while RAN continued to show a large effect size on literacy skills, $F(6, 590) = 18.56$, $p < .001$, $\eta_p^2 = .159$. Both oral-language covariates had a medium to large effect on literacy outcomes, $F(6, 590) = 5.82$, $p < .001$, $\eta_p^2 = .056$, for Receptive morphosyntactic skills, and $F(6, 590) = 15.01$, $p < .001$, $\eta_p^2 = .133$, for Vocabulary. To sum up, children in non-vulnerable schools outperformed children

in vulnerable schools across literacy outcomes, even after controlling for several demographic and contextual factors, as well as for a series of key cognitive skills. The type of school children attended influenced, in particular, higher-order literacy skills, such as text-composition quality and reading comprehension. It was only when considering oral-language factors that School SES ceased to have a significant impact on literacy performance. Notably, children's home SES was generally less critical than School SES; moreover, it was no longer a relevant factor for reading and writing performance when children's cognitive skills and, in particular, their language skills, were included in the analyses.

## Discussion

This paper aimed to examine disparities in literacy learning that may exist between children from different socioeconomic backgrounds. Specifically, our goal was to understand the impact that attending socioeconomically vulnerable schools may have on children's learning and development of reading and writing skills. To accomplish our goal, we recruited a large sample of children attending public schools, which differed greatly in their socioeconomic makeup, and contrasted their reading and writing performances while controlling for several covariates. A major finding of the study was that S-SES had an effect across reading and writing skills, even after controlling for children's individual SES. This finding is well aligned with previous studies, which indicated that the composition of the school, particularly its socioeconomic characteristics, has an impact on academic and, specifically, literacy achievements, above and beyond the effect of familial SES [13]. It should be noted that most previous research on the topic operationalized school SES as the average or aggregation of children's individual SES [9]. In this sense, we have arrived at the same conclusion, but using a different means of classifying educational centers as vulnerable or non-vulnerable, using an external, objective set of criteria. Therefore, the present study provides additional support to the claim that schools with higher proportions of socioeconomically disadvantaged students obtain poorer literacy scores and are at a higher risk of school failure. When C-SES and children's exposure to the language of instruction were considered in the models, the effect of S-SES remained, but it was reduced in size. It also limited its scope of influence only to higher-order literacy skills: the quality of written text productions and reading comprehension, whereas more basic, mechanical aspects of literacy were spared: handwriting, spelling, the amount of text generated, and word reading. Put differently, the negative impact on literacy scores resulting from attending a vulnerable school was ameliorated by children's familial SES and their familiarity with (and, possibly, competence in) the language of instruction. The fact that children coming from households with more highly educated parents, who also tend to have better income levels, cannot elude the negative impact of attending a vulnerable school requires further consideration. Previous literature has indicated that a literacy-rich home environment acts as a facilitator of later literacy development [17]. In part, this may be because such homes are usually those where parents have more intellectual and material resources (including more time) to both be aware of the importance of literacy for their child and to have frequent, high-quality, literacy-related interactions with their children (e.g., joint reading or writing [69]). These households usually present a myriad of opportunities for children to witness literacy practices, enhancing their understanding of the social functions of reading and writing, from a very early age. In consequence, written language, as a domain of knowledge, is not at all foreign to such children when they enter school. In contrast, children from lower-income, low educational-level households will only rarely be exposed to literate practices and may have literacy-related interactions of lesser quality, less often or not at all. As a result, written language may be perceived by them as a skill that is dissociated from their out-of-school experience. All

in all, it is noteworthy that literacy-rich environments were insufficient to compensate for the impact that attending a vulnerable school had on higher-order literacy skills. We would like to speculate that there are at least two possible explanations for this phenomenon: on the one hand, both text writing and reading comprehension are extremely complex skills that have been found to improve only after extended, high-quality, explicit teaching of domain-specific strategies [70, 71]. On the other hand, perhaps the nature of parent-child literacy-related interactions is such that they do not have a powerful effect beyond the mechanical aspects of literacy; in other words, children's higher familiarity with the functions and characteristics of written language may not be enough to facilitate their learning of high-level literacy skills. Finally, when we introduced subject-level covariates that measured cognitive skills, known to underpin both reading and writing, the effect of C-SES was lost; after introducing linguistic-skills covariates, the effect of S-SES was also lost. This finding further substantiates claims for the critical cognitive underpinnings of reading and writing [31, 40]; it also adds to the robust evidence suggesting that oral language skills are essential to literacy development [32, 33]. The finding also means that SES, both at the individual and at the school-level, was not more relevant than a series of domain-general cognitive skills and, especially, than children's individual linguistic performance, to predict achievement in literacy competences. This is good news, as these are all skills that may be acted on from an educational point of view; that is, most of children's cognitive abilities and their linguistic skills respond successfully to treatment and, thus, should help all children achieve adequate literacy levels [72–77]. This is not to say, however, that it is easy or simple to bring vulnerable children's relevant literacy-related skills up to a level that allows them to overcome their initial disadvantages. On the contrary, there are many challenges ahead that make such goal seem out of hand at the moment. First, there is an urgent need for further studies to pinpoint exactly which skills should be targeted and how. Second, it would be essential for highly-effective, evidence-based writing practices to be implemented at schools at large, so that children make substantial progress in key skills. In this sense, research has indicated that the use of evidence-based practices across the curriculum is quite low [75]. Another burning question is the following: if subject-level factors are more important in determining literacy outcomes in 2nd and 4th graders than S-SES, why do we detect these important SES effects at all? One possible answer to this question is that S-SES effects are actually capturing differences on how vulnerable schools operate, as has been suggested elsewhere [25]. This would mean that children in vulnerable schools may perform worse than children in more affluent schools because they are exposed to a worse school climate, or because their teachers might make less emphasis on academic performance, among other differences. Indeed, it has been shown that improving school climate and strengthening academic-performance goals contribute to improving children's academic outcomes [78]. Most importantly, this is clearly an under-researched area and future studies should try to understand the mechanisms through which S-SES impacts literacy outcomes.

## Educational implications and concluding remarks

Our study on the impact of S-SES on literacy outcomes provided evidence that children attending vulnerable schools are typically outperformed by children attending non-vulnerable schools. However, our findings have also shown that SES at the school and at the individual level were not more important than children's literacy-related skills. Moreover, there is a good chance that S-SES effects may reflect worrying discrepancies in how vulnerable and non-vulnerable schools address students' academic needs, although this claim requires further research. Literacy interventions exist that are effective with low-SES or vulnerable populations [79–81]. However, to the best of our knowledge, no literacy intervention has been able to close

the gap between vulnerable and non-vulnerable students; that is, interventions are as effective on different SES populations [82]. Considering the importance of literacy skills for academic, professional, and personal realization, more research is needed on effective literacy intervention that is able to boost disadvantaged children's competences, particularly focusing on early remediation. Finally, our findings suggest that addressing critical skills supporting literacy development (i.e., cognitive, linguistic) might be more effective than programs to compensate poor SES background effects, although perhaps a combination of approaches may prove to be more successful. Future research should compare the relative efficacy of addressing subject-level skills and/or family-based interventions to improve at-risk children's literacy achievements, particularly with the aim of closing gaps in children's performance as a function of socioeconomic factors.

## Acknowledgments

The authors want to thank the schools and children who participated in the study. We are also thankful to our RAs for their support collecting the data and for evaluating some of the children's productions included in this study. Lastly, we want to thank Dr. Gabriel Liberman and Dr. Amir Hefetz for their help with the statistical analyses.

## Author Contributions

**Conceptualization:** Naymé Salas.

**Data curation:** Naymé Salas, Mariona Pascual.

**Formal analysis:** Naymé Salas.

**Funding acquisition:** Naymé Salas.

**Methodology:** Naymé Salas, Mariona Pascual.

**Project administration:** Naymé Salas.

**Supervision:** Naymé Salas.

**Validation:** Mariona Pascual.

**Writing – original draft:** Naymé Salas, Mariona Pascual.

**Writing – review & editing:** Naymé Salas, Mariona Pascual.

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
