## [Decision Letter · Decision Letter 0]

23 Aug 2023

PONE-D-23-22078Impact of school SES on literacy developmentPLOS ONE

Dear Dr. Salas,

Thank you for submitting your manuscript to PLOS ONE. After careful consideration, we feel that it has merit but does not fully meet PLOS ONE’s publication criteria as it currently stands. Therefore, we invite you to submit a revised version of the manuscript that addresses the points raised during the review process.

We look forward to receiving your revised manuscript.

Kind regards,

Jie Wang, Ph.D.

Academic Editor

PLOS ONE

Journal Requirements:

“This research was supported by grants 2015ACUP 00175 (Recercaixa-ACUP program)and PID2019-108791GA-I00 (Ministerio de Ciencia Innovación y Universidades), awarded to N.S. The funders had no role in study design, data collection and analysis, decision to publish, or preparation of the manuscript.”

Additional Editor Comments (if provided):

The authors should pay careful attention to each of the comments below and address the issues raised by the two reviewers.

Reviewers' comments:

Reviewer's Responses to Questions

**Comments to the Author**

1. Is the manuscript technically sound, and do the data support the conclusions?

Reviewer #1: Yes

Reviewer #2: Partly

2. Has the statistical analysis been performed appropriately and rigorously? 

Reviewer #1: Yes

Reviewer #2: Yes

3. Have the authors made all data underlying the findings in their manuscript fully available?

Reviewer #1: Yes

Reviewer #2: Yes

4. Is the manuscript presented in an intelligible fashion and written in standard English?

Reviewer #1: Yes

Reviewer #2: No

5. Review Comments to the Author

Reviewer #1: Thank you for the opportunity to review your paper. I was so excited to see the topic, and I was even more excited when I began to read it. Every question I might have had about why you conducted your study, how it fits within the existing canon and adds to our collective knowledge, how you established your design and conducted your analysis, etc. you answered and answered well. I sincerely tried to find critical points because I personally find peer review feedback more helpful when it is critical and specific (although an 'accept as is' would never be turned down), but I struggled to find anything. I found some minor typos (e.g., a "nor" where I don't think you mean it, some spelling typos, etc.), but overall, I kept thinking this paper was a master class in how to be both succinct and detailed in such a way that everything was really compelling and clear. The data analysis was especially robust, and the findings are impactful. I did find myself asking in your results section if the reason the C-SES was less critical than the S-SES and no longer relevant when children's cognitive and literacy skills were included was precisely because of how predictive poverty is to cognitive skills and language development that they essentially cancelled each other out because of their correlatedness? It does seem you addressed this more explicitly in your discussion section, which was also really well done, in my opinion. I was really impressed with the design and analysis, and more importantly I was made hopeful by the outcomes because of the implications they hold for future research and practice. I thoroughly enjoyed reading this paper.

Reviewer #2: The manuscript is largely well-written and offers interesting data on an urgent filed of inquiry. Some clarifications are necessary:

- "Grade-level expectations" should be defined. What are the grade-level expectations in Catalan schools for grades 2 and 4?

- The rationale for the included assessment tools should be strengthened.

- The conclusion, stating that it is "good news" that children's individual linguistic and cognitive abilities were as important as SES, since these are possible to improve through educational intervention, is hugely over-simplified. Indeed, many language and cognitive functions can be improved. However, if this was easily done through the delivery of mainstream classroom, how come all students don't develop reading and writing according to plan? The authors must be aware that the promotion of many language skills require expert knowledge and techniques not easily included in mainstream education, require theoretical and therapeutic skills not part of the knowledge-base of general teachers, and not easily tailored to the highly diverse needs of the students in a class.

- Please check your text for writing errors, e.g. "tquality", "missigness", "have also shown than", "considerable more research".

6. PLOS authors have the option to publish the peer review history of their article (what does this mean?). If published, this will include your full peer review and any attached files.

Reviewer #1: **Yes: **Karyn A. Allee

Reviewer #2: No

---

## [Author Response · Author response to Decision Letter 0]

27 Oct 2023

Please, note we also submitted these comments as a separate file.

Response to Reviewers

Reviewer #1: Thank you for the opportunity to review your paper. I was so excited to see the topic, and I was even more excited when I began to read it. Every question I might have had about why you conducted your study, how it fits within the existing canon and adds to our collective knowledge, how you established your design and conducted your analysis, etc. you answered and answered well. I sincerely tried to find critical points because I personally find peer review feedback more helpful when it is critical and specific (although an 'accept as is' would never be turned down), but I struggled to find anything. I found some minor typos (e.g., a "nor" where I don't think you mean it, some spelling typos, etc.), but overall, I kept thinking this paper was a master class in how to be both succinct and detailed in such a way that everything was really compelling and clear. The data analysis was especially robust, and the findings are impactful. I did find myself asking in your results section if the reason the C-SES was less critical than the S-SES and no longer relevant when children's cognitive and literacy skills were included was precisely because of how predictive poverty is to cognitive skills and language development that they essentially cancelled each other out because of their correlatedness? It does seem you addressed this more explicitly in your discussion section, which was also really well done, in my opinion. I was really impressed with the design and analysis, and more importantly I was made hopeful by the outcomes because of the implications they hold for future research and practice. I thoroughly enjoyed reading this paper.

>>We are sincerely grateful for Reviewer 1’s comments. We have revised the manuscript entirely and corrected typos and small mistakes. Thank you for your thoughtful comments.

Reviewer #2: The manuscript is largely well-written and offers interesting data on an urgent filed of inquiry. Some clarifications are necessary:

- "Grade-level expectations" should be defined. What are the grade-level expectations in Catalan schools for grades 2 and 4?

>>We have added a reference to the educational Catalan curriculum, which served as the criteria guiding grade-level expectations (please, refer to page 13 of the revised manuscript and reference number 57).

- The rationale for the included assessment tools should be strengthened.

>>We have added an introductory paragraph to our Tasks and measures subsection detailing the rationale for the selection of data-collection instruments. Please, see page 11.

- The conclusion, stating that it is "good news" that children's individual linguistic and cognitive abilities were as important as SES, since these are possible to improve through educational intervention, is hugely over-simplified. Indeed, many language and cognitive functions can be improved. However, if this was easily done through the delivery of mainstream classroom, how come all students don't develop reading and writing according to plan? The authors must be aware that the promotion of many language skills require expert knowledge and techniques not easily included in mainstream education, require theoretical and therapeutic skills not part of the knowledge-base of general teachers, and not easily tailored to the highly diverse needs of the students in a class.

>>We completely agree with this observation by Reviewer 2. Accordingly, we have now expanded our original intention which was to draw attention to the fact that the solution may be implemented in the context of educational centers (as opposed to changes to the social makeup of society, which would clearly be out of our hands), but we now make a subsequent comment about the difficulty of achieving this goal. Please, see page 25.

Please check your text for writing errors, e.g. "tquality", "missigness", "have also shown than", "considerable more research".

>>Done.

---

## [Decision Letter · Decision Letter 1]

27 Nov 2023

Impact of school SES on literacy development

PONE-D-23-22078R1

Dear Dr. Salas,

We’re pleased to inform you that your manuscript has been judged scientifically suitable for publication and will be formally accepted for publication once it meets all outstanding technical requirements.

Kind regards,

Jie Wang, Ph.D.

Academic Editor

PLOS ONE

Additional Editor Comments (optional):

Reviewers' comments:

Reviewer's Responses to Questions

**Comments to the Author**

1. If the authors have adequately addressed your comments raised in a previous round of review and you feel that this manuscript is now acceptable for publication, you may indicate that here to bypass the “Comments to the Author” section, enter your conflict of interest statement in the “Confidential to Editor” section, and submit your "Accept" recommendation.

Reviewer #2: All comments have been addressed

2. Is the manuscript technically sound, and do the data support the conclusions?

Reviewer #2: Yes

3. Has the statistical analysis been performed appropriately and rigorously? 

Reviewer #2: Yes

4. Have the authors made all data underlying the findings in their manuscript fully available?

Reviewer #2: Yes

5. Is the manuscript presented in an intelligible fashion and written in standard English?

Reviewer #2: Yes

6. Review Comments to the Author

Reviewer #2: Thank you for revising the paper in accordance with my recommendations. I look forward to eventually seeing the paper online.

7. PLOS authors have the option to publish the peer review history of their article (what does this mean?). If published, this will include your full peer review and any attached files.

Reviewer #2: No

---

## [Editor Report · Acceptance letter]

4 Dec 2023

PONE-D-23-22078R1 

Impact of school SES on literacy development 

Dear Dr. Salas:

I'm pleased to inform you that your manuscript has been deemed suitable for publication in PLOS ONE. Congratulations! Your manuscript is now with our production department. 

Kind regards, 

on behalf of

Dr. Jie Wang 

Academic Editor

PLOS ONE